# Pointwise Mutual Information Based Metric and Decoding Strategy for Faithful Generation in Document Grounded Dialogs

**Yatin Nandwani, Vineet Kumar, Dinesh Raghu, Sachindra Joshi** and **Luis A. Lastras**

IBM Research, AI

{yatin.nandwani@, vineeku6@in, diraghu1@in, jsachind@in, lastrasl@us}.ibm.com

## Abstract

A major concern in using deep learning based generative models for document-grounded dialogs is the potential generation of responses that are not *faithful* to the underlying document. Existing automated metrics used for evaluating the faithfulness of response with respect to the grounding document measure the degree of similarity between the generated response and the document's content. However, these automated metrics are far from being well aligned with human judgments. Therefore, to improve the measurement of faithfulness, we propose a new metric that utilizes (Conditional) Point-wise Mutual Information (PMI) between the generated response and the source document, conditioned on the dialogue. PMI quantifies the extent to which the document influences the generated response – with a higher PMI indicating a more faithful response. We build upon this idea to create a new decoding technique that incorporates PMI into the response generation process to predict more faithful responses. Our experiments on the BEGIN benchmark demonstrate an improved correlation of our metric with human evaluation. We also show that our decoding technique is effective in generating more faithful responses when compared to standard decoding techniques on a set of publicly available document-grounded dialog datasets.

## 1 Introduction

Document–grounded dialog agents converse with users based on information present in document provided to them. These agents are expected to be factually consistent or *faithful* to the grounding document and refrain from generating content that cannot be verified using the document. As most existing document–grounded dialog agents (Prabhumoye et al., 2021; Wu et al., 2021) are built by fine-tuning large language models, ensuring faithful response generation is a major challenge.

To measure the ability of dialog agents to generate faithful responses, several automatic metrics

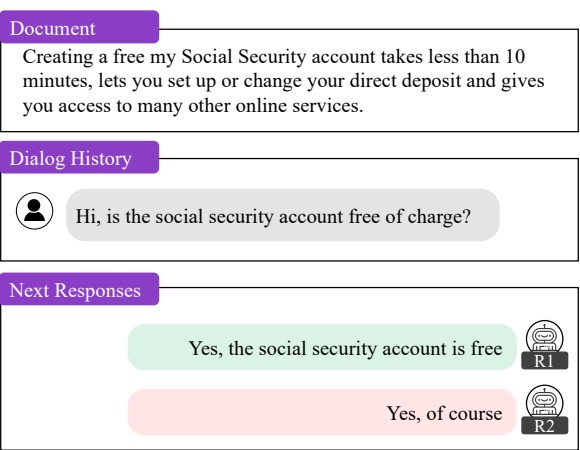

Figure 1: An example document grounded dialog with two types of responses: sentential response (R2) and non-sentential response (R1).

have been proposed. These metrics take as input the agent generated response and the grounding document to quantify faithfulness. These are based on lexical overlap (e.g., BLEU, unigram-F1), semantic overlap (BERTScore) or even a trained classifier (Dziri et al., 2022a). Recently, Honovich et al. (2021) proposed $Q^2$, a metric that measures faithfullness using automatic question generation and question answering.

A major limitation of existing metrics is that they ignore the crucial dialog history when measuring faithfulness of responses. Even though, in many cases, the dialog history provides essential context that is necessary for a complete understanding of the response. To illustrate this point, let's consider two responses, textitR1 and *R2* , as depicted in Figure 1. Response *R1* is self-contained and can be comprehended without relying on the dialog history. On the other hand, response *R2* is dependent on the dialog history and can only be fully understood when considering the preceding conversation. Unfortunately, current automated metrics do not take into account the dialog history, leading to their failure in evaluating responses that are not self-

contained. Responses like $R2$ often lack domain-specific words, making similarity-based metrics like unigram-F1 and BERTScore ineffective. Additionally, generating question-answer pairs using such responses typically captures incomplete information, rendering metrics like Q2 as inadequate measures.

To overcome this problem, we propose a new metric that quantifies the faithfulness of a generated response with respect to both the document and the dialog history. Our metric is grounded in information theoretic concepts and captures the association of the response with the given document using Conditional Pointwise Mutual Information (CPMI). We call our metric PMI–FAITH, which uses CPMI between the generated response and the document, conditioned on the dialogue history, for quantifying faithfulness. PMI–FAITH captures the intuition that for a response to be grounded in the document, the probability of its generation given the document should be higher than the probability of its generation without the document.

A significant advantage of our metric PMI–FAITH is that it can be factorized the same way as the likelihood of a response can be factorized in auto regressive models. We take advantage of this property to propose a novel decoding objective, PMI–DECODE. The goal of PMI–DECODE is to maximize not just the response's likelihood but a score that combines its likelihood and faithfulness.

To summarize, our contributions are threefold:

1. We propose PMI–FAITH, a novel metric which quantifies faithfulness as a conditional PMI between the response and the document given the dialog history.
2. We propose a novel decoding objective, PMI–DECODE, which can aid in generating faithful responses.
3. Our experiments show that PMI–FAITH correlates with human judgments better than any existing metrics on the BEGIN benchmark (Dziri et al., 2022b). We also show that using PMI–DECODE as the objective generates more faithful responses than standard likelihood objective on three standard document–grounded dialog datasets.

We release our code [1] for further use by the research community.

---

[1] https://github.com/ynandwan/pmi-faith

## 2 Related Work

In this work, we focus primarily on faithfulness aspect of the generated responses with respect to the grounding document. It is crucial to distinguish between faithfulness and hallucination (Maynez et al., 2020) in evaluating responses. A response is considered faithful only when all the information it contains can be verified or inferred from the grounded document. On the other hand, a response is considered as a hallucination if it provides false or fabricated information. It is important to note that there can be responses that are not hallucinations but are still unfaithful. In such cases, the information provided may not be false, but it cannot be verified using the grounded document as a reference. It is important to point out that the set of faithful responses is a subset of responses that are not hallucinations. In this section, we discuss related work in faithfulness, followed by a brief discussion on Mutual Information in conversational settings.

Researchers have used various terms such as faithfulness (Cao et al., 2018), factual consistency (Cao et al., 2020; Santhanam et al., 2021), factual accuracy (Goodrich et al., 2019), fidelity (Chen et al., 2020), attribution (Rashkin et al., 2021a) and hallucination (i.e., the lack of faithfulness) (Xiao and Wang, 2021) to define and quantify faithfulness of a model's generated text to a given knowledge.

Most of the works focusing on evaluating faithfulness propose to train a classifier for the task (Goodrich et al., 2019; Kryscinski et al., 2020; Dziri et al., 2022a). Whereas our proposed metric doesn't require any training and is agnostic to the underlying data. Recently, Honovich et al. (2021) proposed $Q^2$ for quantifying faithfulness. It uses a question generator to first generate question–answer (QA) pairs from the generated response. Then a QA system is used to find an answer, to the generated question, from the document. Finally, an NLI system is used to compare the two answers. Though $Q^2$ uses the given document to check the faithfulness of a response, it ignores the dialog history. Thus, it may fail at handling responses that are non-self contained as depicted in Figure 1. Our metric PMI–FAITH addresses this issue.

Many recent works (Dziri et al., 2022b; Honovich et al., 2022) have released different benchmarks that can be used to evaluate the performance of faithfulness metrics. While Honovich et al. aim to standardize benchmark datasets across different

generation tasks, Dziri et al. focus on document–grounded dialogues, and thus we use their benchmark to compare our metric with various baselines.

Li et al. (2016) and Paranjape and Manning (2021) use mutual information to prevent the conversational models from generating generic responses (such as "Sorry, I'm not sure about this topic"). Contemporary to our work, Ren et al. (2023) propose to use a Conditional Pointwise Mutual Information (CPMI) based metric to evaluate relevance of a generated response with respect to a reference hypothesis for open-domain response generation. In contrast, our work is the first to use CPMI as a metric for evaluating the faithfulness of a response given a document and dialogue history in a document-grounded response generation.

## 3 Background

In this section, we first review the task of document-grounded dialog response generation, followed by the definition of the faithfulness metric.

**Document Grounded Response Generation:** Let dialog history $\mathbf{h} = [u_1, \cdots u_m]$ be a sequence of $m$ utterances in the dialog so far and $\mathbf{d}$ be the document on which the next response is grounded. The task of document-grounded dialog response generation is to predict the next response, $\mathbf{r} = \langle r_1 r_2 \ldots r_T \rangle$, one token at a time, given the dialog history $\mathbf{h}$ and the document $\mathbf{d}$. Here, $\forall i, r_i \in \mathcal{V}$, where $\mathcal{V}$ is the vocabulary of all possible tokens. The underlying model learns a probability distribution $P(\mathbf{r}|\mathbf{d}, \mathbf{h})$ over all possible responses $\mathbf{r} \in \mathcal{V}^+$, where $\mathcal{V}^+$ is the space of all the sequences having one or more tokens from vocabulary $\mathcal{V}$.

Typically, this distribution is factorized over the tokens of $\mathbf{r}$ as:

$$P(\mathbf{r}|\mathbf{d}, \mathbf{h}) = \prod_{t=1}^{T} P(r_t|\mathbf{d}, \mathbf{h}, \mathbf{r}_{1:t-1}) \quad (1)$$

**Faithfulness Metric:** Most of the existing definitions (and metrics) for faithfulness focus mainly on document $\mathbf{d}$ and response $\mathbf{r}$ but ignore the history $\mathbf{h}$ (Dziri et al., 2022a,b). This may be for the sake of uniformity across different tasks such as summarization, grounded dialogue generation, and paraphrase generation. We qualify the definition of faithfulness specifically for the task of document–grounded dialogue generation. Formally, a response $\mathbf{r}$ is considered 'faithful' to a given document $\mathbf{d}$ and the dialogue history $\mathbf{h}$ iff $\mathbf{d}, \mathbf{h} \vDash \mathbf{r}$,

where $\vDash$ represents logical entailment.

A faithfulness metric should quantify the faithfulness of the response $\mathbf{r}$ to the document $\mathbf{d}$ and dialogue history $\mathbf{h}$. In general, such a metric should take $\mathbf{r}, \mathbf{d}$ and $\mathbf{h}$ as its input and compute a score, $F(\mathbf{r}, \mathbf{d}, \mathbf{h}) \in \mathbb{R}$, such that a higher value of $F(\mathbf{r}, \mathbf{d}, \mathbf{h})$ indicates a more faithful response.

## 4 Approach

In this section, we describe our proposed metric for faithfulness – PMI–FAITH. We then propose a decoding strategy PMI–DECODE based on our metric, with the objective of generating relevant and faithful responses.

### 4.1 PMI–FAITH

PMI–FAITH is based on the information–theoretic concept of Pointwise Mutual Information. We use the notion of CPMI between generated response $\mathbf{r}$ and the document $\mathbf{d}$ given the context $\mathbf{h}$ to capture the influence of the document in generating the response. We define our metric, PMI–FAITH, for faithfulness of the response $\mathbf{r}$ to the document $\mathbf{d}$ as:

$$\text{PMI–FAITH}(\mathbf{r}, \mathbf{d}, \mathbf{h}) = \text{CPMI}(\mathbf{r}; \mathbf{d}|\mathbf{h})$$
$$= \log \frac{P(\mathbf{r}, \mathbf{d}|\mathbf{h})}{P(\mathbf{r}|\mathbf{h})P(\mathbf{d}|\mathbf{h})} = \log \frac{P(\mathbf{r}|\mathbf{d}, \mathbf{h})}{P(\mathbf{r}|\mathbf{h})} \quad (2)$$

Mathematically, PMI is a measure of the strength of the association between two random events. A positive value of CPMI in eq. (2) implies that the probability of generating the response given the document and the dialogue history is higher than the probability of generating the response given only the dialogue history. Hence, the response is likely to be grounded in the document. On the other hand, if the response $\mathbf{r}$ is not faithful to the document $\mathbf{d}$, the probability of its generation given the document and the dialogue history is likely to be similar to the probability of its generation without the document, resulting in a lower value of PMI–FAITH. We use pre–trained language models such as BLOOM (Scao et al., 2022) or GPT2 (Radford et al., 2019), to compute these conditional probabilities $P(\mathbf{r}|\mathbf{d}, \mathbf{h})$ and $P(\mathbf{r}|\mathbf{h})$.

### 4.2 PMI–DECODE

PMI–DECODE is a decoding strategy whose objective is to generate responses that are both relevant and faithful. Typically, the goal of any decoding

strategy is to select a response that has the maximum (log) likelihood:

$$\mathbf{r} = \arg\max_{\tilde{\mathbf{r}} \in \mathcal{V}^+} \log P(\tilde{\mathbf{r}}|\mathbf{d}, \mathbf{h}) \tag{3}$$

The objective of PMI–DECODE is to select a response that is highly likely and faithful. This is achieved by maximizing a combination of likelihood and faithfulness quantified using an appropriate metric $F$. With $\alpha \in [0, 1]$, and a linear scoring function, we get:

$$\mathbf{r} = \arg\max_{\tilde{\mathbf{r}} \in \mathcal{V}^+}(1 - \alpha) \log P(\tilde{\mathbf{r}}|\mathbf{d}, \mathbf{h}) + \alpha F(\tilde{\mathbf{r}}, \mathbf{d}, \mathbf{h}) \tag{4}$$

With an auto–regressive model that generates the response one token at a time, we use decoding strategies, such as greedy decoding, beam search, nucleus sampling (Holtzman et al., 2020), or beam sampling as a heuristic to find the maxima. For ease of description, we use the greedy decoding below, though our approach is agnostic to the choice of heuristic for maximising the objective function. It just modifies the standard log–likelihood objective with an additional term corresponding to faithfulness. Our choice of PMI–FAITH as function $F$ for quantification of faithfulness keeps the decoding heuristic tractable as shown below.

With eq. (3) as the objective, greedy decoding would sample the next token $r_t$ as follow:

$$\begin{aligned} r_t &= \arg\max_{v \in \mathcal{V}} \log P(\mathbf{r}_{1:t-1}, v|\mathbf{d}, \mathbf{h}) \\ &= \arg\max_{v \in \mathcal{V}}[\log P(\mathbf{r}_{1:t-1}|\mathbf{d}, \mathbf{h}) \\ &\quad + \log P(v|\mathbf{d}, \mathbf{h}, \mathbf{r}_{1:t-1})] \end{aligned} \tag{5}$$

In eq. (5), the likelihood term has been factorized and notice that its first term is independent of the next token candidate $v$ and thus can be dropped while taking $\arg\max$. Not all faithfulness metrics can be decomposed the same way as the likelihood term. One advantage of PMI–FAITH is that it can be decomposed the same way as likelihood as follows:

$$\begin{aligned} &\text{PMI–FAITH}(\mathbf{r}_{1:t-1}, v, \mathbf{d}, \mathbf{h}) \\ &= \log \frac{P(\mathbf{r}_{1:t-1}, v|\mathbf{d}, \mathbf{h})}{P(\mathbf{r}_{1:t-1}, v|\mathbf{h})} \\ &= \log \frac{P(\mathbf{r}_{1:t-1}|\mathbf{d}, \mathbf{h})}{P(\mathbf{r}_{1:t-1}|\mathbf{h})} \\ &\quad + \log \frac{P(v|\mathbf{d}, \mathbf{h}, \mathbf{r}_{1:t-1})}{P(v|\mathbf{h}, \mathbf{r}_{1:t-1})} \\ &= \text{PMI–FAITH}(\mathbf{r}_{1:t-1}, \mathbf{d}, \mathbf{h}) \\ &\quad + \text{CPMI}(v; \mathbf{d}|\mathbf{h}, \mathbf{r}_{1:t-1}) \end{aligned} \tag{6}$$

By using PMI–FAITH as $F$ in eq. (4), and dropping the two terms which are independent of $v$ from eq. (5) and eq. (6), the objective of greedy decoding using the PMI–DECODE objective is expressed as:

$$\begin{aligned} r_t = \arg\max_{v \in \mathcal{V}} &(1 - \alpha) \log P(v|\mathbf{d}, \mathbf{h}, \mathbf{r}_{1:t-1}) \\ &+ \alpha \text{CPMI}(v; \mathbf{d}|\mathbf{h}, \mathbf{r}_{1:t-1}) \end{aligned} \tag{7}$$

To compute CPMI in eq. (7), the same language model can be used to get the conditional probabilities $P(v|\mathbf{d}, \mathbf{h}, \mathbf{r}_{1:t-1})$ and $P(v|\mathbf{h}, \mathbf{r}_{1:t-1})$ by separately passing $\mathbf{d}, \mathbf{h}, \mathbf{r}_{1:t-1}$ and $\mathbf{h}, \mathbf{r}_{1:t-1}$, respectively, through the model.

We observed that using CPMI in the scoring function sometimes results in selecting tokens from the document which may interfere with the grammar. To mitigate this, instead of maximizing over the entire vocabulary $\mathcal{V}$ at each step $t$, we propose to maximize only over the '*top p*' subset from the likelihood distribution, $\mathcal{V}_{p,t}$, defined as the minimum cardinality subset of tokens with the sum of their probabilities as $p$. We call this top $p$ masking:

$$\begin{aligned} r_t = \arg\max_{v \in \mathcal{V}_{p,t}} &(1 - \alpha) \log P(v|\mathbf{d}, \mathbf{h}, \mathbf{r}_{1:t-1}) \\ &+ \alpha \text{CPMI}(v; \mathbf{d}|\mathbf{h}, \mathbf{r}_{1:t-1}) \end{aligned} \tag{8}$$

The intuition here is that while CPMI has a positive influence on generating a more faithful response, it may negatively impact the grammatical structure. Therefore by restricting the vocabulary to $\mathcal{V}_{p,t}$, we use only highly probable tokens to form a response and thus are likely to generate responses that are faithful as well as grammatically correct.

## 5 Experiments

Our experiments answer two research questions:

1. PMI–FAITH: How does our novel metric perform when compared to exisitng metrics on a standard benchmark (section 5.2)?

2. PMI–DECODE: Does our proposed decoding technique generate responses that are more faithful compared to vanilla decoding techniques, while still maintaining relevance (section 5.4).?

## 5.1 PMI–FAITH: Experimental Setup

**Dataset:** We experiment using recently proposed BEGIN benchmark (Dziri et al., 2022b) for evaluating the ability of PMI–FAITH to identify faithful responses. This benchmark uses three document grounded datasets, *viz*, CMU–DoG (Zhou et al., 2018), TopicalChat (Gopalakrishnan et al., 2019), and WoW (Dinan et al., 2019). It contains 11,059 responses generated by three different models, GPT2 (Radford et al., 2019), DoHA (Prabhumoye et al., 2021) and T5 (Raffel et al., 2020), on randomly selected samples from test splits of the 3 datasets. Each generated response is annotated by humans and classified into either 'Fully–attributable', 'Generic', or 'Not fully–attributable'. Overall, 23.3% of the total response have been classified as 'Fully–attributable' by human annotators.

**Evaluation Metrics:** Our objective is to identify faithful responses. Accordingly, we consider 'Fully–attributable' as the positive class and both 'Generic' and 'Not fully–attributable' as the negative class. We use the same evaluation setup as recommended in Honovich et al. (2021). For each metric, we first normalize the score using min-max normalization. The min and the max scores are identified from the dev set. We then identify an optimum threshold for each metric as the one that achieves the best F1 on the dev set. Finally, during test, we use the identified min, max and thresholds to classify a response as faithful. The thresholds, min and max for each metric are reported in Appendix B. Once we have the predicted class, we then compute precision, recall, F1 score and accuracy achieved by each metric. As done in Honovich et al. (2021), we also report calibration-free metrics that don't require any normalization. In addition to Spearman's and Pearson's correlation with human annotations, we also report AUROC for various faithfulness metrics.

**Baselines:** We compare our model against $Q^2$, the state-of-the-art metric. We also compare against various lexical and semantic similarity-based, and trained classifiers as faithfulness metrics. Specifically, we use Unigram-F1 (U-F1), SacreBLEU(Post, 2018; Papineni et al.,

| Metric | Precision | Recall | F1 | Accuracy |
|---|---|---|---|---|
| U-F1 | 0.401 | 0.785 | 0.531 | 0.677 |
| BLEU | 0.478 | 0.479 | 0.479 | 0.757 |
| RougeL | 0.487 | 0.552 | 0.518 | 0.760 |
| BERTScore | 0.459 | 0.673 | 0.546 | 0.739 |
| FaithCritic | **0.684** | 0.492 | 0.573 | 0.829 |
| $Q^2$ | 0.517 | 0.744 | 0.610 | 0.779 |
| UPMI–FAITH | 0.592 | 0.704 | 0.643 | 0.818 |
| PMI–FAITH | 0.607 | **0.818** | **0.697** | **0.834** |

Table 1: Performance of various faithfulness metrics on the BEGIN Benchmark.

2002), and RougeL (Lin, 2004) to capture lexical overlap between **d** and generated response **r**; BERTScore(Zhang et al., 2020) to capture **r**'s semantic similarity with **d**. We use the code[2] provided by Honovich et al. (2021) for all the above baselines.[3] We also compare against *faithcritic*[4](Dziri et al., 2022a), which is a pre–trained classifier to predict faithfulness of a response.

**Training Details:** To measure PMI–FAITH, we need to compute two conditional probabilities: $P(\mathbf{r}|\mathbf{d}, \mathbf{h})$, and $P(\mathbf{r}|\mathbf{h})$. To do so, we use pretrained LLMs available off the shelf from huggingface library (Wolf et al., 2019). To quantify the impact of using one language model over the other, we compute the performance of PMI–FAITH using eight LLMs of varying sizes: five BLOOM (Scao et al., 2022) models with up to 7 billion parameters, and three GPT2 (Radford et al., 2019) models up to GPT2-large (774 million). We observe a robust and consistent performance with a variability of only 0.02 points in the F1 score. Hence, for all further experiments, we use BLOOM-560m.

**Unconditional variant of PMI–Faith:** To quantify the impact of dialogue history **h** on PMI–FAITH, we also use a variant of it called unconditional PMI between a response and a document, *i.e.*, UPMI–FAITH $= \log P(\mathbf{r}|\mathbf{d}) - \log P(\mathbf{r})$, to measure faithfulness.

## 5.2 PMI–FAITH: Experimental Results

Table 1 reports the precision, recall, F1 score, and accuracy achieved by different metrics on the test split of BEGIN benchmark. We first observe that PMI–FAITH performs better than UPMI–FAITH, clearly demonstrating the advantage of using the

---

[2]https://github.com/orhonovich/q-squared

[3]For BERTScore, we change the underlying model to the current best *microsoft/deberta-xlarge-mnli*.

[4]https://huggingface.co/McGill-NLP/roberta-large-faithcritic

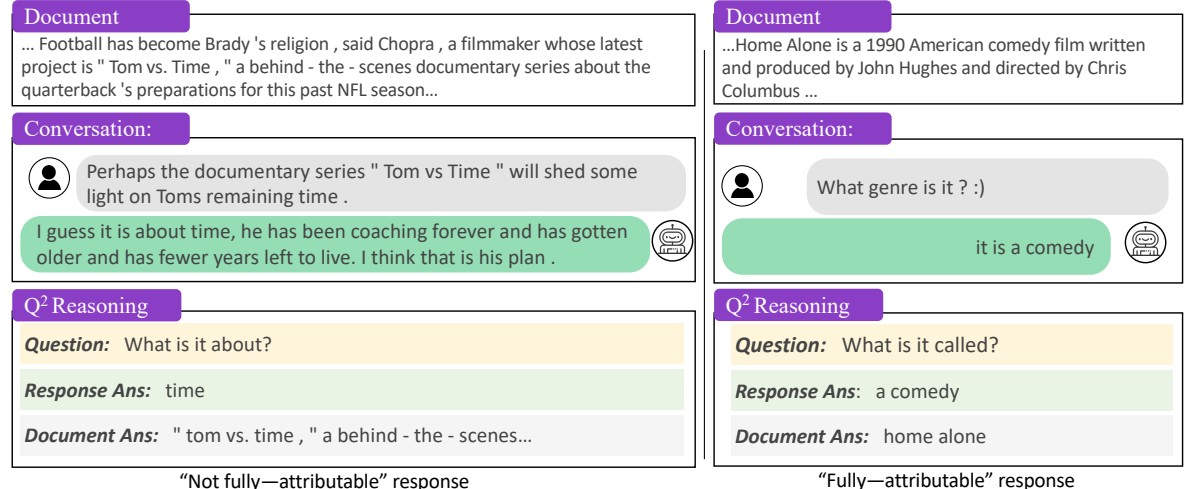

Figure 2: A 'Fully attributable' (right) and a 'Not fully attributable' (left) example, incorrectly classified by $Q^2$ and correctly classified by PMI–FAITH.

| | CMU | TC | WoW |
|---|---|---|---|
| BERTScore | 0.292 | 0.432 | 0.612 |
| FaithCritic | 0.156 | 0.039 | **0.794** |
| $Q^2$ | 0.543 | 0.487 | 0.705 |
| UPMI–FAITH | 0.539 | 0.542 | 0.733 |
| PMI–FAITH | **0.663** | **0.584** | 0.771 |

Table 2: F1 score over each of the three contributing datasets in the BEGIN, *viz.*, CMU–DoG (CMU), TopicalChat (TC), and Wizards of Wikipedia (WoW).

| | Spearman Cor. | Pearson Cor. | AUROC |
|---|---|---|---|
| F1-U | 0.449 | 0.447 | 0.804 |
| BLEU | 0.238 | 0.388 | 0.663 |
| RougeL | 0.412 | 0.465 | 0.781 |
| BERTScore | 0.433 | 0.479 | 0.796 |
| FaithCritic | 0.479 | 0.479 | 0.712 |
| $Q^2$ | 0.447 | 0.448 | 0.795 |
| UPMI–FAITH | 0.528 | 0.485 | 0.861 |
| PMI–FAITH | **0.597** | **0.650** | **0.907** |

Table 3: Spearman and Pearson correlation with human annotations in the BEGIN Benchmark, and AUROC of various faithfulness metrics.

dialogue history while measuring faithfulness.

We then observe that both UPMI–FAITH and PMI–FAITH perform better than all other faithfulness metrics by a considerable margin across all reported performance measures, with the absolute gains ranging from $21.8\%$ to $8.7\%$ in F1 score. Even against the strong baseline of $Q^2$, PMI–FAITH achieves an absolute gain of $5.6\%$ and $8.7\%$ in accuracy and F1 score, respectively. As expected, all the lexical overlap and semantic similarity based metrics achieve poor performance, with accuracy worse than even the majority–class classifier's accuracy of $76.7\%$.

Next, we notice that all metrics, except faithcritic, have higher recall than precision, indicating that they tend to be lenient while classifying a response as faithful, whereas faithcritic tends to be conservative and classifies most of the responses as not faithful. Comparing the next two best metrics, we observe that $Q^2$ has better F1 and recall but worse accuracy and precision than faithcritic.

To identify dataset specific biases, Table 2 reports F1 score separately for each of the three contributing datasets. We observe that PMI–FAITH achieves the highest F1 on CMU–DoG and TopicalChat with more than $12\%$ and $9.7\%$ absolute gain, respectively, over the other metrics. FaithCritic achieves the best F1 score on WoW, whereas its F1 on TopicalChat and CMU–DoG is quite low. This over-fitting on WoW is because faithcritic is a learned metric, and the training data for it has been adapted from WoW, and its low performance on the other two datasets demonstrates its lack of generalization.

To understand the correlation of various faithfulness metrics with human judgement, we report three calibration-free metrics in table 3. In all three metrics, we observe that PMI-Faith is better aligned to human judgements than the other measures of faithfulness.

**Subjective Analysis:** The state-of-the-art metric, $Q^2$, identifies whether a response is faithful or not using two steps. In the first step, it generates a set

of questions based on the response. In the second step, it uses a question answering system to generate two responses for each question: one based on the response and one based on the document. If both the answers match, then the response is considered faithful to the document. We now discuss the shortcoming of $Q^2$ and how PMI–FAITH overcomes it using two examples.

Figure 2 shows the two examples where PMI–FAITH correctly identifies the faithfulness (or lack of it) whereas the strongest baseline $Q^2$ fails to do so. In the case of 'Fully–attributable' response (right), the pronoun '*it*' in the response is an anaphora, referring back to the antecedent 'home alone' (movie name), which is difficult to infer without the dialogue context. However, $Q^2$ doesn't take the dialogue history into account, and thus it considers the pronoun *it* in the response as a cataphor, referring to its postcedent 'comedy'. As a result, the QA system correctly answers the generated question 'What is *it* called?' with the postcedent '*comedy*', when presented with the response, and correctly outputs its antecedent (*home alone*) when presented with the document. But the overall $Q^2$ system fails, as the two answers do not match. On the other hand, by virtue of considering dialogue history during computation, PMI–FAITH has information that the question is about the genre and not the movie name, and hence it can correctly classify the response as 'Fully–attributable'.

The other example highlights two issues: (1) when the response is partially hallucinated, question generation system may generate question from just the faithful part of the response and may incorrectly declare the whole response as faithful. In this example, most of the response contains an opinion, which is not faithful to the document, but the QG system focused on '*I guess it is about time*'. (2): the other issue is that the NLI system fails to capture that the single word answer '*time*' from the response is not entailed by the long answer from the document, resulting in the incorrect prediction by the overall system. On the other hand, PMI–FAITH considers the response as a whole, instead of separately focusing on parts of it. As a result, it is correctly able to identify the given response as not faithful.

## 5.3 PMI–DECODE: Experimental Setup

**Datasets:** We perform our experiments on three document-grounded dialog datasets: Multi-

| Decode Method | Obj. | Faithfulness | | Relevance | |
|---|---|---|---|---|---|
| | | PMI-F | Q2 | BLEU | RougeL |
| MultiDoc2Dial | | | | | |
| Beam Search | Stand. | 0.59 | 0.63 | 30.56 | 0.488 |
| | PMI-D | 0.64 | 0.65 | 28.95 | 0.473 |
| Beam Sampling | Stand. | 0.58 | 0.62 | 30.50 | 0.491 |
| | PMI-D | 0.63 | 0.66 | 30.69 | 0.488 |
| TopicalChat | | | | | |
| Beam Search | Stand. | 0.49 | 0.68 | 6.63 | 0.219 |
| | PMI-D | 0.57 | 0.73 | 5.65 | 0.197 |
| Beam Sampling | Stand. | 0.49 | 0.67 | 6.28 | 0.214 |
| | PMI-D | 0.54 | 0.72 | 6.02 | 0.207 |
| FaithDial | | | | | |
| Beam Search | Stand. | 0.54 | 0.83 | 13.53 | 0.404 |
| | PMI-D | 0.63 | 0.87 | 12.38 | 0.389 |
| Beam Sampling | Stand. | 0.52 | 0.82 | 13.30 | 0.398 |
| | PMI-D | 0.61 | 0.87 | 12.38 | 0.388 |

Table 4: Faithfulness and relevance metrics computed for various decoding techniques on three datasets.

Doc2Dial (Feng et al., 2021), TopicalChat (Gopalakrishnan et al., 2019) and FaithDial (Dziri et al., 2022b). Each dialog in MultiDoc2Dial (MD2D) is between a user and an agent. Only the agent has access to the documents. So, we only model the agent responses for this dataset. TopicalChat (TC) consists of dialogs between two parties, where each party may have a different set of documents on the same topics. We use the 'rare' version of the dataset and filtered utterances tagged as 'personal knowledge'. FaithDial (FD), a faithful adaptation of WoW(Dinan et al., 2019), in which one participant can ask a wide range of questions and the other participant can only provide information from Wikipedia. Some statistics of the three datasets are in Table 6.

**Algorithms:** For each of the three datasets, we separately finetune a BART-Large (Lewis et al., 2019)[5] model using the code[6] made available by Dziri et al. (2022a). As baselines, we use two decoding techniques that use the standard likelihood as the objective function: (1) beam search and (2) beam sampling. Both the techniques use a beam size of 4. We compare these baselines with their variants that uses our PMI–DECODE (PMI–D) objective. We use the values of $\alpha$ and top-p masking that achieved the highest sum of RougeL and normalized PMI–FAITH on the dev set.

[5]https://huggingface.co/facebook/bart-large
[6]https://github.com/McGill-NLP/FaithDial

| | $\alpha$ | PMIF | | | | RougeL | | | | Grammatical errors (in %) | | | |
|---|---|---|---|---|---|---|---|---|---|---|---|---|---|
| $p$ | | **0** | **0.25** | **0.5** | **1** | **0** | **0.25** | **0.5** | **1** | **0** | **0.25** | **0.5** | **1** |
| **0.6** | | 0.52 | 0.58 | 0.59 | 0.59 | 0.40 | 0.40 | 0.39 | 0.38 | 6.9 | 10.2 | 11.8 | 12.2 |
| **0.75** | | 0.52 | 0.60 | 0.61 | 0.60 | 0.40 | 0.39 | 0.38 | 0.36 | 6.4 | 12.0 | 16.3 | 18.6 |
| **0.9** | | 0.52 | 0.61 | 0.62 | 0.57 | 0.40 | 0.39 | 0.36 | 0.30 | 7.5 | 15.0 | 24.2 | 45.8 |
| **1** | | 0.52 | 0.61 | 0.56 | 0.34 | 0.40 | 0.38 | 0.30 | 0.05 | 7.4 | 16.5 | 54.8 | - |

Table 5: Faithfulness, relevance, and grammatical errors of the responses generated by beam sampling using different configurations of $\alpha$ and $top-p$ masking on the FaithDial dataset

.

| | Num. samples | | | Avg. words | | |
|---|---|---|---|---|---|---|
| | **Train** | **Dev.** | **Test** | **Doc.** | **Hist.** | **Resp.** |
| **MD2D** | 24,603 | 4,699 | 4,567 | 166 | 93 | 18 |
| **TC** | 131,555 | 8,183 | 8,301 | 241 | 199 | 20 |
| **FD** | 18,357 | 3,417 | 3,539 | 23 | 69 | 18 |

Table 6: Various statistics of the three document grounded dialog datasets.

| | Multi-Doc2Dial | | | Topical Chat | | |
|---|---|---|---|---|---|---|
| | Fai | Rel | Gra | Fai | Rel | Gra |
| Standard | 0.52 | 0.72 | 0.96 | 0.69 | 0.70 | 0.96 |
| PMI-D | 0.60 | 0.75 | 0.92 | 0.80 | 0.67 | 0.93 |

Table 7: Human evaluation of responses generated using beam search with different decoding objectives. We evaluate faithfulness (Fai), relevance (Rel) and grammar (Gra).

## 5.4 PMI–DECODE: Experimental Results

We compare the decoding strategies in Table 4 using various automated metrics for faithfulness (PMI–FAITH, $Q^2$) and relevance (BLEU, RougeL). The general trend is that the PMI–DECODE generates more faithful responses compared to the standard variant and the improvement in faithfulness comes a the cost of relevance.

To gain a better understanding of the correlation between faithfulness and relevance, we conducted experiments involving different configurations of the control parameters of PMI–D ($\alpha$ and top-p masking). The results of these experiments are presented in the table 5, showcasing the corresponding normalized PMIF and RougeL metrics for each configuration.

For a given fixed value of $\alpha$, as the $p$ value increases, the faithfulness of the responses also increases. However, this improvement in faithfulness comes at the cost of decreased relevance. On the other hand, for a fixed $p$ value, as $\alpha$ increases, the faithfulness initially increases and then gradually decreases. Simultaneously, the relevance decreases with an increase in $\alpha$. The highest level of faithfulness is observed when $\alpha = 0.5$ and $p = 0.6$. Meanwhile, the highest relevance is achieved when the CPMI is not used in the decoding objective ($\alpha = 0$), or when both $\alpha$ and $p$ have low values.

We have gained two valuable insights from table 5. The first insight reveals that there are specific configurations, such as $p = 0.6$ and $\alpha = 0.25$, which achieve the same level of relevance (0.40) as the standard variant while generating more faithful responses. The second insight highlights a significant drop in relevance when solely focusing on PMI scores ($\alpha = 1$ and $p = 1$). Therefore, if the goal is to generate responses with nearly equivalent relevance, utilising smaller values for $\alpha$ along with a masking value of 0.6 is recommended.

To demonstrate the impact of top-p masking on the grammar, we also report the percentage of grammatically incorrect responses for different values of $\alpha$ and top-p masking in table 5. We use GECToR (Omelianchuk et al., 2020), a grammatical error correction method, to find if a generated response is grammatically correct or not. We can easily see that for any value of $\alpha > 0$, the grammatical errors reduce significantly with a reduction in $top-p$. We do not report any value for $\alpha = 1$ and $top-p = 1$ as the responses with this configuration are not even in proper English. For example, one of the responses generated with $\alpha = 1$ and $top-p = 1$ is "*Cheyne Lauren sisters Vel Lauren wear Ralph indo Austrian Ralph linesauxricting Ren therapies Combat Rarity glamorous*".

**Human Evaluation:** We perform human evaluation experiments to compare (1) *relevance*, (2) *faithfulness*, and (3) *grammar*. All three dimensions were categorically labeled as agree, neutral, or disagree. We sampled 100 random (document, dialog history, response) tuples, 50 each from MD2D and TC. We evaluate the responses generated by beam search using two objectives:

| | |
|---|---|
| **Document** | **What happens if I had a lapse of insurance coverage during active duty?** You are required to maintain continuous coverage or surrender your plates before deployed. If you were unable to do so , DMV has a procedure in place to exempt you from administrative actions upon your return. You must provide a notarized statement that the vehicle was not used during the time in question, a copy of your military ID, proof of current insurance or surrender of plates, and signed and dated copies of deployment and return papers, or DD-214 if separated from service... |
| **Context** | **User**: lost my insurance while on active duty and have some questions
**Agent**: You are required to maintain continuous coverage or surrender your plates before being deployed. Were you unable to keep your insurance while on active duty?
**User**: yes, it just wasn't on my mind and I don't get notices and anything like that while deployed |
| **Greedy** | **Agent**: Unfortunately, no relevant information is found. |
| **PMI**: | **Agent**: DMV has a procedure in place to exempt you from administrative actions upon your return. You must provide a notarized statement that the vehicle was not used during the time in question, a copy of your military ID , proof of current insurance or surrender of plates, and signed and dated copies of deployment and return papers , or DD-214 if separated from service. |

Table 8: An example from the test set of Multi-Doc2Dial dataset where Greedy generates a '*I don't know*' response and PMI–DECODE generates a relevant and faithful response.

standard and PMI–DECODE. Out of six in–house annotators used (3 per dataset), four were experts in dialog research and two were beginners. Refer to appendix A for more details.

The results are summarized in Table 7. For each dimension, we report the percentage of responses that were rated *agree*. As expected, PMI–DECODE generates more faithful responses compared to greedy. We observe a 15% improvement in faithfulness compared to greedy decode on both datasets. Further, PMI–DECODE improves relevance on MD2D but slightly deteriorates on TC. Manual analysis revealed that the improvement of relevance on MD2D is primarily due to inherent solution multiplicity (Nandwani et al., 2021) in most dialogues, where more than one correct response is possible, but the metrics capture just one.

As PMI–DECODE maximises not just the likelihood of responses, but a combination of likelihood and faithfulness, we expected the responses to contain grammatical errors compared to greedy decode. To counter this issue, we proposed to use a weighted combination of likelihood and faithfulness during decode, with a higher weight on likelihood. We also restricted the vocabulary during each decode step to just the *top-p* subset. The human study shows that these mitigation techniques helped in reducing the grammatical mistakes made by PMI–DECODE. We see that the grammar is only slightly inferior to greedy on both the datasets.

We use Fleiss Kappa (Fleiss and Cohen, 1973) to measure the inter-annotator agreement, which is substantial for relevance (0.63) and faithfulness (0.63), and almost perfect (0.88) for grammar.

**Subjective Analysis:** Table 8 presents an example from MD2D where standard likelihood based beam search decoding returns a generic response ('*no info. found*') which is present in around 1800 training samples. The same model returns the correct response when PMI–DECODE objective is used instead of just likelihood, demonstrating the capability of PMI–DECODE to shift the score in favour of the words present in the document.

## 6   Conclusion

In this paper, we present a novel metric, PMI–FAITH, to measure faithfulness of responses generated by document grounded dialog systems. It uses conditional PMI between the response and the document given the dialog history to quantify faithfulness. We extend the idea of PMI–FAITHto propose a novel decoding objective, PMI–DECODE which encourages responses to be faithful to the given document by maximizing both the likelihood and faithfulness of the decoded response. Our experiments on the BEGIN benchmark prove that our proposed metric better correlates with human judgments compared to existing metrics. On three document-grounded dialog datasets, our novel decoding objective generates more faithful responses than the standard likelihood objective, as measured using automated metrics and a human study.

## Limitations

Though our decoding objective generates more faithful responses, we observed its inability to respond to generic chit–chat or pleasantries, like 'Hello!' or 'Good–bye'. It is possible to combine it

with other techniques, like training with CTRL tokens (Rashkin et al., 2021b), which can enable it to generate both generic as well as faithful responses depending upon the dialogue context. But identifying when to generate a particular kind of response may require more insights and we leave this overall thread for future work. Next, to compute CPMI, we need to pass **d**, **h**, and **h** separately to the decoder. Though it can be done in parallel, but it may still reduce the throughput of the overall system by half. Finally, as demonstrated by the human evaluation, PMI–DECODE at times generates grammatically incorrect responses, even though the pre–trained language models are very good at generating fluent and coherent English. While we presented two knobs: $\alpha$ and $top\ p$ masking to overcome this, we believe there could be other ways of handling this.

## Ethics Statement

Our work does not introduce any new ethical concerns per se, other than the ones already faced by large language models. Our decoding objective works on top of any trained language model and generates the text which is more faithful to a given input document. This can act as a double–edged sword: on one hand, if the document itself contains profanity, it may enhance the model's likelihood of generating similar content. But on the other hand, providing a valid document may also reduce the inherent likelihood of the model to generate profane content. Therefore, we recommend using it with responsibility and caution.

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

## A   Human Evaluation

The screenshot of a sample task is shown in Figure 3.

For our human evaluation study, we ensured the quality and expertise of our annotators by selecting individuals who are fluent in English and have a solid foundation in Machine Learning (ML), and Natural Language Processing (NLP). Out of six in–house annotators used, four were experts in dialog research and two were beginners. Each annotator had completed at least one formal course in ML/NLP. The exact qualifications and experience of the six annotators are given below:

- Annotator-1 is a postgraduate degree holder with more than 2 decades of experience in NLP research.

- Annotator-2 and annotator-3 are PhD degree holders with more than 5 years of experience in AI.

- Annotator-4 is an undergraduate degree holder with more than 5 years of experience in NLP research.

- Annotator-5 and annotator-6 are undergraduate degree holders with about 2 years of experience in AI research.

All the annotators provided their consent over an appropriate official communication channel, *e.g.*, official email or slack channels.

The following were the instructions provided to the human evaluators.

**What is the task?** There are 50 incomplete dialogs along with a document over which the dialog is grounded on. For each (document, incomplete dialog) pair we provide the next response predicted by 2 different dialog systems (shuffled in random order). You are requested to judge the response generated by these 2 systems along three dimensions: faithfulness, relevance and grammar. Each dimension has to be evaluated using the following scale: Agree (A), Neutral (N), and Disagree (D).

**How to judge relevance?** Relevance measures how apt is the response given the dialog context and the knowledge. Please select *agree* when the response is apt and does not convey any incorrect information. Select *neutral* when it is hard to decide whether it is right or wrong and *disagree* otherwise.

**How to judge faithfulness?** The faithfulness of a response is only dependent on the grounding document and it is independent of the dialog. A system response can be marked disagree for relevance and still be marked agree for faithfulness. Please select *agree* when the complete response can be inferred from the document. Select *neutral* when it is hard to decide whether it can be inferred from the document or not and *disagree* when a major portion of

Figure 3: An example of the task provided to the human judges.

the response cannot be inferred from the document. For the case where the response is something like "No information is present". The judgement should be *agree* if there is no information about that in the document provided and "disagree" if there is information available in the document, but the system didn't pick it up. For cases where the user initiates a chit-chat (say the user says "hi, how are you"), the agent responds with chit-chat ("I am doing good"), please can mark faithfulness as *neutral*.

**How to judge grammar?** The grammar score for a response is independent of the dialog or the document. A system response can be marked as disagree for relevance and still be marked agree for grammar. Please select *agree* when the response looks like how an expert human writes. Select *neutral* when there is a major issue with how the response reads but it still understandable and *disagree* when the response makes no sense.

## B    Faithfulness Metric Normalization

The minimum and maximum values used for normalizing various metrics are shown in Table 9. These are the minimum and maximum values achieved by each metric on the dev set. We also report the threshold used on the normalized metrics for the faithfulness classification task. These are the thresholds than achieved the hightest F1 on the dev set.

|  | Min | Max | Threshold |
|---|---|---|---|
| U-F1 | 0 | 1 | 0.070 |
| BLEU | 0 | 100 | 0.039 |
| RougeL | 0 | 1 | 0.202 |
| BERTScore | -0.639 | 1 | 0.440 |
| FaithCritic | 0 | 1 | 1 |
| $Q^2$ | 0 | 1 | 0.625 |
| UPMI–FAITH | -8.092 | 4.286 | 0.759 |
| PMI–FAITH | -2.069 | 4.334 | 0.534 |

Table 9: . The min and max values used for normalizing each metric and the threshold used for the classifying faithfulness of a response using the metric.