# OpenReview forum: "Pointwise Mutual Information Based Metric and Decoding Strategy for Faithful Generation in Document Grounded Dialogs"
_EMNLP/2023/Conference — EMNLP 2023 Main_

### Official Review · Reviewer_FY11 · 2023-08-01

**Soundness:** 4

**Excitement:**

4: Strong: This paper deepens the understanding of some phenomenon or lowers the barriers to an existing research direction.

**Paper Topic And Main Contributions:**

- Faithfulness metric for the response generation, considering dialogue history. Proposal of the state-of-the art metric, PMI–FAITH.
- Faithfulness-considering decoding strategy, PMI–FAITH, and significant enhancement on the BEGIN benchmark

**Questions For The Authors:**

- I am not sure if "dialogue history" is the real matter as you have noted in the Introduction. According to the proposed methodology, the information disparity derived by the "document" seems to be the key aspect in guaging faithfulness. Have you done any additional analysis on the importance of considering dialogue history? Although dialogue history is considered in the proposed CPMI, it seems that CPMI can be implemented without dialogue history, and it seems that this can also produce quite meaningful results.

- In equation 2, were P(r|d, h) and P(r|h) estimated by different models? I find that this paper utilized fine-tuned model for such probabilities, but I wonder that whether you intended to use respective model or not. If different, P(r|d, h) and P(r|h) would be better to be noted like P(r|d, h, \theta) and P(r|h, \phi).

- In equation 2, I find that p(r|h) can be vanished to 0 and may yield CPMI explosion. Is there any cases in practical use concerning it?

- In equation 5, it seems that r_t = arg max_{v∈V} log P(v|r1:t−1,d, h) in the first stage (paper described r_t = arg max_{v∈V} log P(r1:t−1, v|d, h)). The generation probability of the t^th token is grounded at "1:t-1"^th tokens, and it seems that the generation probability of "1:t-1" is not considered together. If wrong, please let me know.

- In implementing PMI–DECODE based inference, it seems that lagging will inevitably occur. How much is the speed difference from normal decoding?

**Reasons To Accept:**

- Problem statement and aiming objective are very clearly described: distinction between faithfulness and hallucination
- New state-of-the art performance metric for estimating faithfulness of generated dialogue
- Novel decoding strategy for enhancing faithfulness in dialogue generation

**Reasons To Reject:**

Nothing special, but I wonder the following questions.

**Reproducibility:**

4: Could mostly reproduce the results, but there may be some variation because of sample variance or minor variations in their interpretation of the protocol or method.

**Reviewer Confidence:**

4: Quite sure. I tried to check the important points carefully. It's unlikely, though conceivable, that I missed something that should affect my ratings.

**Typos Grammar Style And Presentation Improvements:**

- line 58: textitR1
- line 587: PMI–FAITHto

---

> ### Author Rebuttal · Authors · 2023-08-29
>
> Dear reviewer FY11,
>
> We thank you for a careful review of our work and for asking pertinent questions. We appreciate that you found our problem formulation clear and liked the clear distinction between faithfulness and hallucination in the related works.  We are glad that you didn’t find any special reason to reject our work. Below we try our best to answer your questions.
>
> **Question-A:** *I am not sure if "dialogue history" is the real matter as you have noted in the Introduction. According to the proposed methodology, the information disparity derived by the "document" seems to be the key aspect in gauging faithfulness. Have you done any additional analysis on the importance of considering dialogue history? Although dialogue history is considered in the proposed CPMI, it seems that CPMI can be implemented without dialogue history, and it seems that this can also produce quite meaningful results.*
>
> You are absolutely right in pointing out that PMI-Faith aims to capture the information disparity derived by the document.  We note that none of the existing metrics exploit dialogue history to quantify faithfulness and hence the emphasis in the introduction.
>
> We condition on dialogue history while computing the conditional probabilities as it is crucial for a good performance. We thank you for suggesting an alternative PMI based metric that doesn’t condition on dialogue history to demonstrate this aspect. As suggested, we use PMI, defined as  $log (P(r | d) / P(r) )$, as an additional metric for quantifying faithfulness and call it Unconditional-PMI-Faith (U-PMI-F). We observe that it indeed performs slightly inferior to PMIFaith (in terms of F1) and better than all other metrics. The tables below are a replica of Table 1 and Table 2 with an additional row for U-PMI-F:
>
> Table 1: Performance of various faithfulness metrics on the BEGIN Benchmark
>
> |    **Metric**   | **Precision** | **Recall** |    **F1** | **Accuracy** |
> |:---------------:|--------------:|-----------:|----------:|-------------:|
> | **FaithCritic** | **68.4%**     | 49.2%      | 57.3%     | 82.9%        |
> |      **Q2**     | 51.7%         | 74.4%      | 61.0%     | 77.9%        |
> | **U-PMI-F** | 59.2%         | 70.4%      | 64.3%     | 81.8%        |
> |    **PMI-F**    | 60.7%         | **81.8%**  | **69.7%** | **83.4%**    |
>
> Table 2: F1 score over each of the three contributing datasets in the BEGIN, viz., CMU–DoG (CMU), TopicalChat (TC), and Wizards of Wikipedia (WoW).
>
> | **Metric**      | **CMU**   | **TC**    | **WoW**   |
> |-----------------|-----------|-----------|-----------|
> | **FaithCritic** | 15.6%     | 3.9%      | **79.4%** |
> | **Q2**          | 54.3%     | 48.7%     | 70.5%     |
> | **U-PMI-F** | 53.9%     | 54.2%     | 73.3%     |
> | **PMI-F**       | **66.3%** | **58.4%** | 77.1%     |
>
> **Question-B:** *In equation 2, were P(r|d, h) and P(r|h) estimated by different models? I find that this paper utilized fine-tuned model for such probabilities, but I wonder that whether you intended to use respective model or not. If different, P(r|d, h) and P(r|h) would be better to be noted like P(r|d, h, \theta) and P(r|h, \phi).*
>
> Both $P(r | d, h)$ and $P(r | h)$ were estimated using the same model and hence we dropped dependence on model parameters for conciseness.
>
> In the experiments, we used finetuned models only for generating responses in section 5.3 (PMI–DECODE: Experimental Setup), but not for computing PMI-Faith.
>
> **Question-C:** *In equation 2, I find that p(r|h) can be vanished to 0 and may yield CPMI explosion. Is there any cases in practical use concerning it?*
>
> Theoretically, it may explode. But practically, we never observed this when computing PMI-Faith. Since we are evaluating responses generated by a language model, and computing the probabilities also using a language model, albeit a different one, we do not observe $P( r | h)$ close to 0 for the generated responses.
>
> On the other hand, in PMI-Decode, a very low value of $P(v | h, r_{1:t-1})$ may result in an arbitrary token getting selected and this is one of the issues that our masking strategy elegantly prevents from happening.
>
> **Question-D:** *In equation 5, it seems that r_t = arg max_{v∈V} log P(v|r1:t−1,d, h) in the first stage (paper described r_t = arg max_{v∈V} log P(r1:t−1, v|d, h)). The generation probability of the t^th token is grounded at "1:t-1"^th tokens, and it seems that the generation probability of "1:t-1" is not considered together. If wrong, please let me know.*
>
> Your understanding is absolutely correct and both the equations are equivalent.  Consider:
>
>  $r_t = arg max_{v∈V} log P(r_{1:t−1}, v|d, h))$
>
> Now applying Bayes rule, we can represent $P(r_{1:t-1} , v | d, h )$ in the RHS as $P(r_{1:t-1} | d, h) * P(v | r_{1:t-1}, d, h)$. We get:
>
> $r_t = arg max_{v∈V} ( log P(r_{1:t-1} | d, h)  + log P(v | r_{1:t-1}, d, h) )$
>
> Since $P(r_{1:t-1} | d, h)$ is independent of $v$, it can be dropped from RHS, resulting in exactly what you described:
>
> $r_t = arg max_{v∈V} (log P(v | r_{1:t-1}, d, h) )$
>
> **Question-E:** *In implementing PMI–DECODE based inference, it seems that lagging will inevitably occur. How much is the speed difference from normal decoding?*
>
> We need to compute both $P(r | d, h)$, and $P (r | h)$ at every step of the decoder. In our implementation, we compute both these probabilities in parallel on the GPU by batching them together in the same request. Specifically, we make use of the LogitProcessor class in HuggingFace to combine the probabilities at every decode step. In our experiments, we observed that PMI Decode with beam size 4 takes 429 ms per example whereas standard beam search takes 312 ms per example on a A100 GB GPU.

---

### Official Review · Reviewer_RmAJ · 2023-08-03

**Paper Topic And Main Contributions:** 1) They propose a metric that utilize…
**Typos Grammar Style And Presentation Improvements:** 1. line # 58, please apply textit sty…
**Soundness:** 3

**Excitement:**

3: Ambivalent: It has merits (e.g., it reports state-of-the-art results, the idea is nice), but there are key weaknesses (e.g., it describes incremental work), and it can significantly benefit from another round of revision. However, I won't object to accepting it if my co-reviewers champion it.

**Missing References:**

[1] C-PMI: Conditional Pointwise Mutual Information for Turn-level Dialogue Evaluation
- I know it is published after the submission. But I think you should include the paper as well in the review process just in case for the acceptance.

[2] Human-like informative conversations: Better acknowledgements using conditional mutual information



**Questions For The Authors:**

A. Please indicate the V+ in line 180 (Section 3). It's quite confusing while reading without a clear explanation.
B. Is there any clear reason why you compare the evaluation metrics with the classification task by deciding the threshold? Is there any benchmark in comparing evaluation metrics? Why didn't you normalize the score directly from the raw evaluation results?
C. I understand that the Q^2 metric is SOTA, but I think it would be better to show the examples which got incorrect with Faithcritic but correct with your metric as well in subjective analysis.
D. The p-masking was proposed to mitigate grammar issues, but the experimental results in Table 5 only show the relevance and faithfulness differences. I would recommend adding grammar-related metrics to show its improvements.
E. Who were the human evaluators? graduate students? undergraduate? fluent in English? I need more details.

**Reasons To Accept:**

1) Their suggestion involves a measurement which makes use of Conditional Point-wise Mutual Information (C-PMI) between the response produced and the source document. This measurement is influenced by the ongoing conversation and aims to assess faithfulness.

2) They develop a decoding method that integrates PMI into the process of generating responses. This integration enhances the ability to anticipate and generate responses that are more faithful to the source document.

**Reasons To Reject:**

A. I'm not sure the evaluation setting is validated. (Please refer to Question B). Why did you evaluate generation metrics with a classification setting? Please give us some benchmark settings.
B. Human evaluation needs more details to validate the PMI-D's performance. WoW dataset is removed from the human evaluation while WoW is included in automatic evaluation.
C. I think the related works that PMI is adapted in a conversational setting to indicate the differences.

**Reproducibility:**

4: Could mostly reproduce the results, but there may be some variation because of sample variance or minor variations in their interpretation of the protocol or method.

**Reviewer Confidence:**

4: Quite sure. I tried to check the important points carefully. It's unlikely, though conceivable, that I missed something that should affect my ratings.

---

> ### Author Rebuttal · Authors · 2023-08-29
>
> Dear Reviewer RmAJ,
>
> Thank you for a detailed review of our manuscript and for pointing out missing references and typos.  We will include the missing references in the camera ready version.  We also thank you for suggesting additional evaluations, specially automated evaluation of grammar, that will further strengthen our paper. We address your concerns and answer your questions below:
>
> **Reason-to-Reject-1:** *I'm not sure the evaluation setting is validated. Is there any clear reason why you compare the evaluation metrics with the classification task by deciding the threshold? Is there any benchmark in comparing evaluation metrics? Why didn't you normalize the score directly from the raw evaluation results?*
>
> Thank you for raising this concern. As suggested, we have added a few more standard ways for comparing the evaluation metrics below.
>
> Our evaluation setting of using classification task for comparing faithfulness metrics is same as the “Response-Level Evaluation” proposed by Q2 (Honovich et. al., 2021). We used a threshold as Honovich et. al. suggested the following in Section 5.1 of their paper - “*We note that the threshold was arbitrarily selected for the purpose of demonstrating Q2’s ability in separating consistent from inconsistent content, and properly tuning it by splitting the data into development and test sets may improve the results further.*” So, we selected an optimal threshold for all the metrics using the small dev set provided in the BEGIN benchmark.
>
> We use BEGIN bechmark which is a standard dataset for comparing various faithfulness metrics.  Each sample in the BEGIN benchmark has a model-generated response that has been annotated for faithfulness by *humans*. As our goal is to design a metric that mimics human behaviour, we measure how close are the existing and proposed metric to human judgement of faithfulness. In the same way as Q2 (Table 5 in Honovich et. al., 2021), we report three calibration-free metrics to quantify the alignment of various metrics with human judgements. In all 3 metrics, we see that PMI-Faith is better aligned to human judgements than the existing metrics.
>
> | **Metric**        | **F1-U** | **BLEU** | **RougeL** | **BERTScore** | **FaithCritic** | **Q2** | **PMI-Faith** |
> |-------------------|:--------:|:--------:|:----------:|:-------------:|:---------------:|:------:|:---------:|
> | **Spearman Cor.** |   0.449  |   0.238  |    0.412   |     0.433     |      0.479      |  0.447 |   0.597   |
> | **Pearson Cor.**  |   0.447  |   0.388  |    0.465   |     0.479     |      0.479      |  0.448 |   0.650   |
> | **AUROC**         |   0.804  |   0.663  |    0.781   |     0.796     |      0.712      |  0.795 |   0.907   |
>
> **Reason-to-Reject-2:**  *Human evaluation needs more details to validate the PMI-D's performance. WoW dataset is removed from the human evaluation while WoW is included in automatic evaluation.*
>
> For our human evaluation study, we ensured the quality and expertise of our annotators by selecting individuals who are fluent in English and have a solid foundation in Machine Learning (ML), and Natural Language Processing (NLP). Each annotator had completed at least one formal course in ML/NLP. The exact qualifications and experience of the six annotators are given below:
> - annotator-1 is a postgraduate degree holder with more than 2 decades of experience in NLP research.
> - annotator-2 and annotator-3 are PhD degree holders with more than 5 years of experience in AI.
> - annotator-4 is an undergraduate degree holder with more than 5 years of experience in NLP research.
> - annotator-5 and annotator-6 are undergraduate degree holders with about 2 years of experience in AI research.
>
> Should the reviewer require more specific details, we are more than willing to furnish them with the necessary information.
>
> As we used our in-house conversational AI experts for the experiments, we were only able to get it annotated for 2 datasets due to resource constraints and hence we randomly chose two datasets for this experiment.
>
> **Reason-to-Reject-3:**  *I think the related works that PMI is adapted in a conversational setting to indicate the differences.*
>
> We thank the reviewer for pointing us to missing related and contemporary works. We will add the missing references in the updated manuscript as given below.
>
> Mutual information has been previously used in conversational settings. Mutual information has been used to prevent the conversational models from generating generic responses (such as “Sorry, I’m not sure about this topic”) [1,2]. Contemporary to our work, Liliang et al (2023) propose to use a Conditional Pointwise Mutual Information (C-PMI) based metric to evaluate **relevance** of a generated response with respect to a reference hypothesis for open-domain response generation. In contrast, our work is the first to use C-PMI as a metric for evaluating **faithfulness** of a response given a document and dialogue history in a document-grounded response generation.
>
> [1] Ren, Liliang, et al. "C-PMI: Conditional Pointwise Mutual Information for Turn-level Dialogue Evaluation." Proceedings of the Third DialDoc Workshop on Document-grounded Dialogue and Conversational Question Answering. 2023.
>
> [2] Li, Jiwei, et al. "A Diversity-Promoting Objective Function for Neural Conversation Models." Proceedings of the 2016 Conference of the North American Chapter of the Association for Computational Linguistics: Human Language Technologies. 2016.
>
> [3] Paranjape, Ashwin, and Christopher D. Manning. "Human-like informative conversations: Better acknowledgements using conditional mutual information." Proceedings of the 2021 Conference of the North American Chapter of the Association for Computational Linguistics: Human Language Technologies. 2021.
>
> **Question-A:** *Please indicate the V+ in line 180 (Section 3). It's quite confusing while reading without a clear explanation.*
>
> We apologize for the confusion caused. $V^+$ refers to the space of all the sentences having one or more token from vocabulary $V$. We will clarify this in the updated manuscript.
>
> **Question-B:** *Is there any clear reason why you compare the evaluation metrics with the classification task by deciding the threshold? Is there any benchmark in comparing evaluation metrics? Why didn't you normalize the score directly from the raw evaluation results?*
>
> Please refer to Reason-to-Reject-1
>
> **Question-C:** *I understand that the Q^2 metric is SOTA, but I think it would be better to show the examples which got incorrect with Faithcritic but correct with your metric as well in subjective analysis.*
>
> Some examples where Faithcritic is  incorrect and PMI-Faith is correct.
>
> *Example-1*
>
> - Document: ...Home Alone is a 1990 American comedy film written and produced by John Hughes and directed by Chris Columbus ...
>
> - Dialog History: User: What genre is it ? :)
>
> - Response: it is a comedy
>
> - Gold Label: Faithful
>
> *Example-2*
>
> - Document:  …American football, referred to as football in the United States and Canada, and also known as gridiron football or simply gridiron, is a team sport played by two teams of eleven players on a rectangular field with goalposts at each end….
>
> - Dialog History: User: How many players are on the field at once?
>
> - Response: 12 players .
>
> - Gold Label: Not faithful
>
> **Question-D:** *The p-masking was proposed to mitigate grammar issues, but the experimental results in Table 5 only show the relevance and faithfulness differences. I would recommend adding grammar-related metrics to show its improvements.*
>
> Thank you for suggesting this evaluation. Following your comment, we used GECToR [1], a grammatical error correction method, to find if the response is grammatically correct or not. The 2D grid below contains the percentage of grammatically incorrect responses for different values of $\alpha$ and $top-p$ mask. We can easily see that for any value of $\alpha > 0$, the grammatical errors reduce significantly with reduction in $top-p$.
>
>
> | **$\alpha$** |     **0**    | **0.25** | **0.5** | **1** |
> |-------------|--------------|----------|---------|-------|
> |   **top-p**   |         |      |     |   |
> |   **0.6**   |     6.9%     |   10.2%  |  11.8%  | 12.2% |
> |   **0.75**  |     6.4%     |   12.0%  |  16.3%  | 18.6% |
> |   **0.9**   |     7.5%     |   15.0%  |  24.2%  | 45.8% |
> |    **1**    |     7.4%     |   16.5%  |  54.8%  |   -   |
>
> We do not report any value for $\alpha = 1$ and $top-p = 1$ as the responses there are not even in proper English.  An example response from the $\alpha = 1$ and $top-p = 1$ setup is shown below:
>
> “Cheyne Lauren sisters Vel Lauren wear Ralph indo Austrian Ralph linesauxricting Ren therapies Combat Rarity glamorous”
>
> [1] Omelianchuk, Kostiantyn, et al. "GECToR–Grammatical Error Correction: Tag, Not Rewrite." Proceedings of the Fifteenth Workshop on Innovative Use of NLP for Building Educational Applications. 2020.
>
> **Question-E:** *Who were the human evaluators? graduate students? undergraduate? fluent in English? I need more details.*
>
> Please refer to Reason-to-Reject-2.

---

### Official Review · Reviewer_AFXM · 2023-08-04

**Soundness:** 4

**Excitement:**

4: Strong: This paper deepens the understanding of some phenomenon or lowers the barriers to an existing research direction.

**Paper Topic And Main Contributions:**

The authors proposed Point-wise Mutual Information (PMI) between the generated response and the source document to measure the faithfulness of response with respect to the grounding document considering the dialogue history.
In addition, they proposed a new decoding strategy based on the PMI to generate more faithful responses.


**Reasons To Accept:**

The authors not only demonstrated improved performance of the proposed method, but also provided detailed analyses in comparison with state-of-the-art methods. It appears that the proposed metric and the decoding strategy based on this metric are effective for evaluating factuality in dialogue. PMI-FAITH can also be easily computed using only a pre-trained model, which implies the wide acceptability of this method. Finally, it was proven that PMI-D has superior faithfulness according to human evaluation.

**Reasons To Reject:**

There is no report on human evaluation comparing PMI-FAITH to other metrics. It seems that additional proof is required to substantiate the effectiveness of this metric and the author's argument that previous automated metrics are far from being well aligned with human judgments.

**Reproducibility:**

2: Would be hard pressed to reproduce the results. The contribution depends on data that are simply not available outside the author's institution or consortium; not enough details are provided.

**Reviewer Confidence:**

3: Pretty sure, but there's a chance I missed something. Although I have a good feel for this area in general, I did not carefully check the paper's details, e.g., the math, experimental design, or novelty.

---

> ### Author Rebuttal · Authors · 2023-08-29
>
> Dear reviewer AFXM,
>
> Thank you for a positive review. We are glad that you found our detailed comparison with s-o-t-a useful.  Below we attempt to address your concern regarding the lack of human evaluation of PMI-Faith. In case we have misunderstood your concern, please let us know and we will be happy to clarify.
>
> **Reason-to-Reject-1:** *There is no report on human evaluation comparing PMI-FAITH to other metrics. It seems that additional proof is required to substantiate the effectiveness of this metric and the author's argument that previous automated metrics are far from being well aligned with human judgments.*
>
> We are indeed evaluating PMI-FAITH and other metrics using human annotations. For this, we use BEGIN benchmark (Dziri et. al., 2022). Each sample in this benchmark has a model-generated response that has been annotated for faithfulness by *humans*. As our goal is to design a metric that mimics human behaviour, we measure how correlated are the existing and proposed metrics to the human judgement. In the paper (Table 1), we show that PMI-Faith is the best metric as it matches with human judgement 83.4% times (accuracy).
>
> We have also computed Spearman correlation, Pearson correlation and area under ROC curve (AUROC) for various metrics. This shows that PMI-Faith has the highest correlation with human judgements.
>
> | **Metric**        | **F1-U** | **BLEU** | **RougeL** | **BERTScore** | **FaithCritic** | **Q2** | **PMI-Faith** |
> |-------------------|:--------:|:--------:|:----------:|:-------------:|:---------------:|:------:|:---------:|
> | **Spearman Cor.** |   0.449  |   0.238  |    0.412   |     0.433     |      0.479      |  0.447 |   **0.597**   |
> | **Pearson Cor.**  |   0.447  |   0.388  |    0.465   |     0.479     |      0.479      |  0.448 |  **0.650**   |
> | **AUROC**         |   0.804  |   0.663  |    0.781   |     0.796     |      0.712      |  0.795 |   **0.907**   |
>
> We will add this table to our revised manuscript

---

### Meta-Review · Area_Chair_Ngwd · 2023-10-04

**Recommendation:** 5

**Metareview:**

A Point-wise Mutual Information-based technique to measure and improve the faithfulness of generated response with the grounding document and dialogue history. The paper is well written. Please add addressed feedback in the final version.

---

### Decision · Program_Chairs · 2023-10-07

**Decision:**

Accept-Main

**Comment:**

A Point-wise Mutual Information-based technique to measure and improve the faithfulness of generated response with the grounding document and dialogue history. The paper is well written. Please add addressed feedback in the final version.